# The Contributions of Sub-Communities to the Assembly Process and Ecological Mechanisms of Bacterial Communities along the Cotton Soil–Root Continuum Niche Gradient

**DOI:** 10.3390/microorganisms12050869

**Published:** 2024-04-26

**Authors:** Shaodong Liu, Ruihua Liu, Siping Zhang, Qian Shen, Jing Chen, Huijuan Ma, Changwei Ge, Lidong Hao, Jinshan Zhang, Shubing Shi, Chaoyou Pang

**Affiliations:** 1Engineering Research Centre of Cotton, Ministry of Education/College of Agriculture, Xinjiang Agricultural University, Urumqi 830052, China; 2National Key Laboratory of Cotton Bio-Breeding and Integrated Utilization, Institute of Cotton Research, Chinese Academy of Agricultural Sciences, Anyang 455000, China; 3Zhengzhou Research Base, National Key Laboratory of Cotton Bio-Breeding and Integrated Utilization, Zhengzhou University, Zhengzhou 450001, China; 4Postdoctoral Mobile Station, Lanzhou University, Lanzhou 730000, China

**Keywords:** soil bacterial community, soil–root compartment niches, abundance sub-community, bacterial community assembly, stochastic processes

## Abstract

Soil microbes are crucial in shaping the root-associated microbial communities. In this study, we analyzed the effect of the soil–root niche gradient on the diversity, composition, and assembly of the bacterial community and co-occurrence network of two cotton varieties. The results revealed that the bacterial communities in cotton soil–root compartment niches exhibited a skewed species abundance distribution, dominated by abundant taxa showing a strong spatial specificity. The assembly processes of the rhizosphere bacterial communities were mainly driven by stochastic processes, dominated by the enrichment pattern and supplemented by the depletion pattern to recruit bacteria from the bulk soil, resulting in a more stable bacterial community. The assembly processes of the endosphere bacterial communities were determined by processes dominated by the depletion pattern and supplemented by the enrichment pattern to recruit species from the rhizosphere, resulting in a decrease in the stability and complexity of the community co-occurrence network. The compartment niche shaped the diversity of the bacterial communities, and the cotton variety genotype was an important source of diversity in bacterial communities within the compartment niche. We suggest that the moderate taxa contribute to significantly more changes in the diversity of the bacterial community than the rare and abundant taxa during the succession of bacterial communities in the cotton root–soil continuum.

## 1. Introduction

Soil microbes are important regulators of plant productivity and diversity in terrestrial ecosystems, which includes mutualists, pathogens, and commensals. They are considered to be the “seed bank” of plant root microbes and the main driving force for shaping root-associated microbial communities [1,2,3]. Root-associated microbes include rhizosphere and endosphere microbiota, which colonize the surrounding and internal root tissues of plants, respectively, and affect the growth and development of the host through neutral, beneficial, or detrimental interactions [4]. The rhizosphere is an important interface for plant–soil–microbe interactions and signal transduction, as well as energy and mass exchange [2]. About 20–40% of the photosynthetically fixed carbon in plant roots is utilized by exudates and root cap sediments [5,6], which, together with the plant immune system, regulate the diversity and richness of the root microbial community [7,8]. Rhizosphere microorganisms can expand a plant’s habitable range and metabolic capacity, promote plant growth, improve the availability of nutrients, inhibit plant pathogens, and enhance the ability of plants to resist abiotic stress [9,10,11]. The endosphere is a more independent and exclusive compartment, and endophytic bacteria are mostly derived from the rhizosphere as a sub-community of the rhizobacterial communities [12].

The pronounced microbial separation between compartment niches shows a spatial gradient from the bulk soil to the rhizosphere and then to the endosphere, confirming that rhizo-compartmentalization is the largest source of variation within root-associated microbiomes, as observed in rice [13], *Arabidopsis* [3] and mature poplar trees [14]. Based on previous bacterial community differentiation studies, a “two-step selection model” for root microbiota differentiation has been proposed to reveal the assembly process of microbial communities from the bulk soil to the endosphere [4,15]. First, rhizodeposition drives microbes from the bulk soil to the rhizosphere and forms the rhizosphere microbiota. Second, the rhizosphere microbes enter the endosphere under the regulation of the host genotype and establish the endosphere microbiota [3,4,15]. Factors such as host genotypes, climate (e.g., drought and rainfall), the physicochemical properties of the soil (e.g., soil pH and nutrient content), and human disturbance (e.g., agricultural management regimes) drive the assembly of the plant microbial communities, while the plant host exerts powerful selective effects on the microbes through the immune system and root exudates secretions [1,3,4,16].

The ecological process of the assembly of the microbial community can be explained by a deterministic process based on niche theory and a stochastic process based on neutral theory [17]. The assembly of the plant microbiome begins shortly after seeding and is driven by a combination of deterministic and stochastic processes [15]. Niche colonization occurs in different habitats under a combined effect of host and environmental factors [15,18]. The microbial communities in most niches usually exhibit a skewed species abundance distribution, in which a few abundant taxa dominate the community, while a large number of rare taxa exist with very low abundances and are ignored [19]. However, recent studies have shown that the rare taxa contain a large number of metabolically active microbial species that help maintain community persistence and stability, play an important role in driving the ecosystem function, and respond to environmental changes differently from more abundant taxa [20,21]. Therefore, these two processes have different effects on rare and abundant taxa, with rare taxa being dominated by deterministic processes and abundant taxa being largely influenced by stochastic processes [22,23]. Additionally, some studies have found that the dominant and rare populations are not fixed, and, with changes in environmental conditions, new dominant populations can emerge from rare populations. As such, the rare biosphere is considered a potentially inexhaustible repository of genomic innovations [24]. The assembly and stability of plant-associated microbiomes are affected by positive, negative, and neutral microbial interactions that form complex networks. Therefore, microbial community interactions in different niches can be characterized by a symbiotic network analysis [25,26]. A co-occurrence network analysis has revealed that non-random co-occurrence patterns are a ubiquitous feature of soil microbes; however, the network topology changes with the environmental conditions [25,27]. Moreover, there has been an increase in research to unravel the ecological mechanisms underpinning the microbiome assembly and the plant–microbiome interactions, which can provide a mechanistic framework to discern these complex interactions within the plant–microbiome holobiont [28].

Cotton (*Gossypium hirsutum* L.) is an important cash crop that provides raw material to the textile industry. Owing to the development of cultivation techniques such as “drip irrigation under mulch”, “short, dense, and early”, and the “integration of water and fertilizer”, the high yield or super high yield of cotton has been ensured. With the implementation of light and simplified cotton farming technology, China’s cotton production has been geared towards sustained, high yields and high efficiency technology [29,30]. Cotton microbiome studies primarily focus on the rhizosphere microbial composition and soil-borne diseases [11,31,32,33]. However, the contribution of abundant sub-communities to the succession of microbial communities in commensal niches has not been systematically analyzed in terms of the composition and structural evolution of the soil–root continuum microbiome.

Allelochemicals secreted by the roots of crops in continuous cropping systems cause nutritional imbalances and autotoxicity [5]. The selective enrichment of soil microorganisms, especially soil-borne pathogenic microorganisms, disrupts the balance between beneficial and harmful microorganisms and significantly changes the diversity and composition of the soil microbial communities [34,35]. Root-associated microbiomes vary by soil and host genotype, and, although the microbial community assembly patterns in the soil–root continuum are complicated and remain controversial [36], there have been few studies on the ecological niches of the cotton soil–root continuum. Therefore, the assembly progress, diversity, composition, function, and co-occurrence network analysis of the bacterial communities within and among the soil- and root-associated microbiomes along compartment niches in situ remain unclear [36].

To address these issues, two cotton varieties, XLZ61 and XLZ62, which are mainly cultivated in the northwest inland cotton region of China, were selected as the research models, and cotton fields where continuous cropping has been performed for more than 10 years were selected to study the microbiome of the soil–root continuum. We studied the bacterial communities in two cotton varieties and three niche gradients (the bulk soil, rhizosphere, and endosphere). Our primary objectives were to (i) understand the effects of the compartment niche and cotton genotypic differences in the diversity and composition of bacterial communities in the soil–root continuum of cotton fields; (ii) clarify the assembly mechanism and ecological processes of the microbial community in the soil–root continuum using a symbiotic network analysis to reveal the internal mechanism of the microbial interactions in the process of community succession; and (iii) investigate the contribution of factors such as the various abundant sub-communities and the variety of genotypes in driving the bacterial community succession, ecosystem function, and community stability.

## 2. Materials and Methods

### 2.1. Field Experimental Design and Sample Collection

The field experiment for this study was conducted in 2019 at the Hu Yanghe Ecological Experiment Station of the Institute of Cotton Research, Chinese Academy of Agricultural Sciences (84.8032 N; 44.7309 E). XLZ61 and XLZ62 were selected for the experiment, which are the most commonly planted varieties in Xinjiang (China). The seeds were sown on 24 April. The soil was collected at the initial flowering stage on 2 July using the S-shaped random sampling method [35]. The sampling tools were disinfected before use, and the sampling points were identified. The soil samples were collected using a 15 cm soil core sampler, at a distance of approximately 20 cm from the base of the plant. The bulk soil samples were collected from the unplanted cotton soil, and the rhizosphere samples were collected 1–2 mm away from the root system. An ethanol-sterilized nylon-bristled toothbrush was used to remove the closely adhering soil from the root, which we collected as the rhizosphere fraction.

The roots were then rinsed with deionized water, patted dry, and the lateral roots were divided into small segments (~5 cm) using sterile scissors, followed by the vortexing of the roots twice for 1 min in an epiphyte removal buffer (ice cold 0.75% KH_2_PO_4_, 0.95% K_2_HPO_4_, and 1% Triton X-100 in ddH_2_O, with the filter sterilized at 0.2 μM). The roots were again rinsed with the clean epiphyte removal buffer and patted dry. All the samples of 5 g were collected and transferred to sterile centrifuge tubes, marked, and stored in liquid nitrogen tanks in the laboratory at −80 °C.

### 2.2. Extraction of Genome DNA and Amplicon Generation

The total genomic DNA was extracted from the samples using the CTAB/SDS method. The DNA concentration and purity were monitored using 1% agarose gel. According to the concentration, the DNA was diluted to 1 ng/μL using sterile water. The 16S rRNA gene Pair-End (PE) amplicon sequencing on the V5 + V7 regions using the primers 799F (5′-AACMGGATTAGATACCCKG-3′) and 1193R (5′-ACGTCATCCCCACCTTCC-3′) was performed on the microbiome DNA samples. The specific primer with the barcode was used to amplify the 16S/18S rRNA genes. 

### 2.3. PCR Products Quantification and Qualification 

The same volume of 1× loading buffer (containing SYB green) was mixed with the PCR products, and an electrophoresis with 2% agarose gel was performed for detection. The samples that generated a bright, broad strip between 400 and 450 bp were chosen for further experiments. The PCR products were mixed in equidense ratios. Then, the PCR product mixture was purified using the GeneJET Gel Extraction Kit (Thermo Scientific, Waltham, MA, USA).

### 2.4. Library Preparation and Sequencing

The sequencing libraries were generated using the NEB Next^®^ Ultra™ DNA Library Prep Kit for Illumina (NEB, Ipswich, MA, USA) following the manufacturer’s recommendations with index codes added. The library quality was assessed by the Qubit@ 2.0 Fluorometer (Thermo Scientific) and Agilent Bioanalyzer 2100 systems. Finally, the library was sequenced on an Illumina HiSeq platform and the 250 bp paired-end reads were generated.

### 2.5. Definition of Abundant Taxa, Moderate Taxa, and Rare Taxa

We modified a previous definition [20,23] by classifying all the OTUs. Here, the OTUs with relative abundances above 1% of the total sequences were defined as “abundant taxa”, those with relative abundances below 0.01% were defined as “rare taxa”, and those with relative abundances between 0.01% and 1% were defined as “moderate taxa” [20,37]. Additionally, we defined the rare taxa and moderate taxa as “low abundance taxa”.

### 2.6. Calculations and Statistical Analyses

The paired-end reads from the original DNA fragments were merged using FLASH (http://www.cbcb.umd.edu/software/flash, accessed on 16 April 2024) [38], which is designed to merge paired-end reads when there are overlaps between reads 1 and 2. The paired-end reads were assigned to each sample according to the unique barcodes. The sequences were analyzed using the Quantitative Insights Into Microbial Ecology (QIIME) [39] software package (Version 1.9.1). There is a certain proportion of dirty data in the raw data obtained by sequencing. In order to make the results of the information analysis more accurate and reliable, the original data are first spliced and filtered to obtain clean data. The clustering of the OTUs (Operational Taxonomic Units) and a species classification analysis were conducted based on the valid data. The sequences with ≥97% similarity were assigned to the same OTUs. We chose representative sequences for each OTU and used the RDP classifier [40] to annotate the taxonomic information for each representative sequence.

The significance of different factors on the microbial community and functional guilds was tested with PERMANOVA or nested PERMANOVA (the crop species nested within the fertilization treatment) using the “adonis” function of the “vegan” package (Version 1.15-1) [41].

The Sloan neutral community model [42] was selected to determine the contribution of stochastic processes to the assembly of the microbial community by predicting the relationship between the frequency with which the taxa occur in a set of local communities (the proportion of local communities in which each taxon is detected) and their abundance in the metacommunity (estimated by the mean relative abundance across all local communities within biomes or clusters). We used the fit of the neutral model (R^2^) to infer the stochastic processes. The *m* value conveys the estimated migration rate; higher *m* values indicate that microbial communities have less limited dispersal [43,44]. All computations were performed in R (version 3.2.3) [45].

The “niche breadth” approach was used to quantify the habitat specialization [43]. The formula is described below:$$B_{j}=\frac{1}{\sum_{i=1}^{N}{\left(P_{ij}\right)}^{2}}$$
where *B_j_* is the niche breadth, *P_ij_* is the proportion of OTU*j* in a given site *i*, and *N* is the total number of sites. A higher *B* value of OTUs indicates that the OTUs were present and more evenly distributed on a large scale. In contrast, lower *B* values indicate that the OTUs occurred in fewer habitats and were unevenly distributed [46]. We calculated the *B* value based on species abundance and evenness for the bulk soil, rhizosphere, and endosphere. 

To determine the relative importance of neutral processes (i.e., dispersal limitation and ecological drift) and niche-based processes in maintaining the microbial communities in the rhizosphere and bulk soil, the data were fitted to the Sloan neutral community model [42]. This model assumes that all OTUs are equivalent, and the migration rate is the only parameter that determines the model fitting to predict the frequency of occurrence using the average abundance of each OTU. The neutral model was fitted using custom R scripts reported previously [44].

### 2.7. Co-Occurrence Network Analysis

To estimate the pattern of species coexistence, metacommunity co-occurrence networks consisting of the niche gradients and varieties were constructed. These were used to calculate the pairwise Spearman’s rank correlations (r) within the “hmisc” R package version 3.12-2. Only robust (r > 0.6 or r < −0.6) and statistically significant (*p* < 0.01) correlations were used in the network analysis. Subsequently, node-level topological properties, including degree, betweenness, closeness, and eigenvector, were calculated in the “igraph” R package version 1.5.0. The values of these topological features reflect the roles of nodes in the network. A set of network-level topological features, such as average degree, connectome, diameter, average path length, and clustering coefficient, were also calculated. The robustness of a network was tested using the natural connectivity of a complex network. The networks were visualized based on Gephi v. 0.9.2, and a modular analysis was performed to research the inner-network structure between the two seasons.

## 3. Results

### 3.1. Differential Analysis of Bacterial Community Composition and Structure along the Cotton Soil–Root Continuum Compartment Niches

In this study, we analyzed the composition of the bacterial community and the structure of three microhabitats along the cotton soil–root continuum compartment niches in Xinjiang and identified significant differences between the microhabitats. Based on the relative abundance of the operational taxonomic units (OTUs), we further classified the bacterial communities into three sub-communities: abundant taxa (AT), moderate taxa (MT), and rare taxa (RT). A total of 1428 OTUs were identified in the bulk soil (BS) bacterial communities, including 23 abundant taxa OTUs with a cumulative relative abundance of 66.31%, and 1058 rare taxa OTUs with a cumulative relative abundance of 3.43%. In the rhizosphere (RS) bacterial communities of the two cotton genotypes (XLZ61 and XLZ62), 1645 and 1660 OTUs were identified, respectively, where the OTU number and the cumulative relative abundance of the moderate taxa were significantly higher than those in the bulk soils (2.91 and 2.71 times the OTU number, and 2.62 and 2.48 times the cumulative relative abundance, respectively, as shown in Table 1). An analysis of the endosphere assisted in identifying 787 and 611 OTUs, respectively, in which the cumulative relative abundance of the abundant taxa reached 64.43% and 59.74%, but the OTU number and the cumulative relative abundance of the moderate and rare taxa were greatly decreased (Table 1).

We used the bulk soil bacterial community as the control treatment and produced a volcanic plot map of the changes in the relative abundance in OTUs between the rhizosphere bacterial communities of the XLZ61 and XLZ62 (Figure 1a). The assembly process of the rhizosphere bacterial community was found to be dominated mainly by the enrichment pattern and supplemented by the depletion pattern to recruit bacteria from the bulk soil. The relative abundance of 856 (XLZ61) and 826 (XLZ62) OTUs significantly increased in the rhizosphere of the two cotton varieties (*p* < 0.05), mainly due to the recruitment of moderate and rare taxa from the bulk soil communities. We also found 248 and 245 emerging OTUs in the rhizosphere environment, which were mainly recruited as moderate and rare taxa in the rhizosphere communities. The relative abundance of the OTUs in the depletion patterns of the two varieties was approximately four and three OTUs, respectively, which represented a significant decrease (*p* < 0.05). Meanwhile, we found that all the abundant taxa in the bulk soil were recruited to the rhizosphere in the depletion pattern, and the cotton rhizosphere re-recruited the unique abundant taxa (Appendix A). Then, a volcanic plot of the changes in the relative abundance of the OTUs of the endophytic bacterial community in the roots of the cotton was produced using the rhizosphere bacterial community of XLZ61 and XLZ62 as the control (Figure 1b). The analysis revealed that the cotton endospheres were dominated by the depletion pattern and supplemented by the enrichment pattern to recruit species from the rhizosphere bacterial community. The relative abundance of 159 and 78 endosphere OTUs increased significantly in the two cotton varieties, respectively, (*p* < 0.05), and 857 and 823 OTUs differed significantly (*p* < 0.05). Meanwhile, 932 and 1110 OTUs were selectively filtered by cotton roots, which were dominated by moderate and rare taxa. Similarly, all abundant taxa species in the rhizosphere of the two cotton varieties were recruited by the root endophytes in the depletion pattern, and the cotton root endophytes re-recruited the unique abundant taxa species (Appendix A).

We also analyzed the bacterial community composition in different compartment niches of the cotton soil–root continuum (Figure 1c). It was revealed that, at the phylum level, the dominant species in the bacterial community of the bulk soil mainly included *Proteobacteria*, *Actinobacteria*, and *Firmicutes*. *Bacteroidetes*, *Acidobacteria*, and *Gemmatimonadetes* were enriched in the rhizosphere, and *Proteobacteria* were enriched in the root endophytic environment. At the class level, we found that *Gammaproteobacteria*, *Deltaproteobacteria*, and *Alphaproteobacteria* of the *Proteobacteria* phylum were specifically enriched in the bulk soil, rhizosphere, and endosphere, respectively. At the genus level, the bulk soil bacterial community was dominated by pathogenic bacteria including *Ralstonia*, *Cutibacterium*, and unidentified *Corynebacteriaceae*. Antagonistic probiotics such as *Pseudomonas*, *Bacillus*, *Arthrobacter*, and unidentified *Acidobacteria* were enriched in the rhizosphere. In contrast, the dominant species in the root endophytic environment included nitrogen-fixing and antibacterial probiotics such as *Altererythrobacter*, *Sphingomonas*, *Ilumatobacter*, and unidentified *Rhizobiaceae*. We also found that *Sphingomonas* maintained relatively high abundance in all three ecological niches, but different strains (OTUs) were significantly enriched in different ecological niches, showing a strong spatial specificity (Appendix A). 

Further analysis of the composition of different sub-communities revealed that the composition of the dominant species in the abundant taxa was generally consistent with the total community, aside from the moderate and rare taxa, which were substantially different. An analysis of the moderate taxa in the endosphere environment revealed that the dominant species were *Gemmatimonadetes*, *Acidobacteria*, and *Bacteroidetes*, as they were significantly enriched. Other present taxa included *Alphaproteobacteria*, *Gammaproteobacteria,* unidentified *Gemmatimonadetes*, unidentified *Acidobacteria*, and *Deltaproteobacteria* at the class level; and *Sphingomonas*, *Arthrobacter*, *Bacillus*, *Candidatus Entotheonella*, *Haliangium*, and unidentified *Acidobacteria* at the genus level, which were also heavily enriched in the root endophytic environment at the phylum level. At the phylum level in the rhizosphere, *Alphaproteobacteria*, *Gammaproteobacteria*, and *Acidimicrobiia* were largely enriched, while, at the genus level, unidentified *Rhizobiaceae*, *Devosia*, *Sphingobium*, *Ramlibacter*, *Mesorhizobium*, and *Bradyrhizobium* were largely enriched. This study of the rare taxa revealed that the species types and relative abundance of the rare taxa were relatively consistent in both the bulk soil and the rhizosphere, but the relative abundance of most dominant species entering the endosphere environment was significantly decreased, and some species failed to access the root system completely (Appendix A).

### 3.2. Differences in Bacterial Community Diversity along the Niche Gradient in the Cotton Soil–Root Continuum

In this study, a Wilcoxon rank-sum test analysis showed that the compartment niche was the main factor affecting the α-diversity of the bacterial communities (Figure 2A). The Shannon index and Chao1 index of the rhizosphere soil bacterial community of Xinjiang cotton varieties were significantly higher than those of the bulk soil and endosphere bacterial communities (*p* < 0.01), while the Chao1 index of the endosphere bacterial community was the lowest and was significantly lower than that of the bulk soils (*p* < 0.01). The difference in cotton genotypes primarily affected the richness of the endosphere bacterial community (Figure 2B). The Chao1 index of the endosphere bacterial community in the XLZ61 was significantly higher than that in the XLZ62 (*p* < 0.05). The Wilcoxon rank-sum test analysis also showed that the variation patterns of the α-diversity of the abundant sub-communities were significantly different between the three niche gradients (Figure 2C,D). The α-diversity index of the abundant taxa (AT) was the lowest among the three abundance sub-communities, and the Chao1 index of the rhizosphere bacterial community between the different niches was the lowest, both of which were significantly lower than those of the bulk soil (*p* < 0.05). The α-diversity index of the moderate taxa (MT) significantly increased and then decreased along the niche gradient (*p* < 0.01), which was consistent with that of the total community. The α-diversity index of the rare taxa species decreased significantly along the niche gradient (*p* < 0.05), which was significantly affected by the host selection. Our analysis showed that in the different sub-communities in the rhizosphere and endosphere environments, the α-diversity indices of the two cultivars differed significantly, especially regarding the rare species (Figure 2E,F). This indicated that host genotype differences had a more significant impact on rare taxa, particularly in the endosphere environment.

In this study, based on the Bray–Curtis distance algorithm, the unconstrained principal coordinate analysis (PCoA) of the bacterial communities was performed, and the principal components with the largest contribution rate were displayed in a graph to assess the differences among microbial communities and to quantify community similarities across compartment niches [47,48]. The analysis showed that the ecological niche gradient was the main factor affecting the separation of the bacterial community structure in the soil–root continuum (Permanova analysis: R^2^ = 0.82; *p* = 0.001), where PCo1 and PCo2 explained 56.44% and 25.98% of the source of community differences, respectively (Figure 3A). The niche gradient had a significant effect on the structure of the three sub-communities (AT: R^2^ = 0.83, *p* = 0.001; MT: R^2^ = 0.73, *p* = 0.001; and RT: R^2^ = 0.39, *p* = 0.001), where PCo1 and PCo2 accounted for 54.39% and 28.01% of the differences in the abundance taxa, 50.35% and 23.03% of the differences in the moderate taxa, and 23.96% and 15.60% of the differences in the rare taxa, respectively (Figure 3B). The genotypic differences between the cotton varieties resulted in a significant separation of the bacterial communities on the first principal component in the rhizosphere (Permanova analysis R^2^ = 0.54, *p* = 0.034) and the endosphere (Permanova analysis R^2^ = 0.4, *p* = 0.03), which explained the source of the differences between the rhizosphere and endosphere bacterial communities, accounting for 55.52% and 42.25%, respectively (Figure 3C). The host genotypes were an important contributor to the structural differences in the rhizosphere and endosphere, as the MT (RS: R^2^ = 0.70, *p* = 0.035 and ES: R^2^ = 0.52, *p* = 0.035) and rare taxa (RS: R^2^ = 0.41, *p* = 0.024 and ES: R^2^ = 0.28, *p* = 0.03) were significantly separated at the first principal component, with the genotype differences explaining the differences in the moderate taxa of the rhizosphere and endosphere of 71.40% and 53.60% and the differences in the rare taxa of 41.54% and 28.72%, respectively (Figure 3D,E).

### 3.3. Assembly Mechanisms of Microbial Communities along the Niche Gradient in the Soil–Root Continuum

We found that the niche breadth index in the rhizosphere was significantly higher than that in the bulk soil and the endosphere (*p* < 0.05). However, there was no significant difference between varieties within the niches (Figure 4A). There were significant differences between the sub-communities within the niche breadth indices (Appendix A). The variety in genotype was the main reason for the significant difference in the abundant taxa niche breadth indices (*p* < 0.05), but the niche gradient was the main reason for the differences in the moderate taxa (*p* < 0.05), which were similar to the total community along the niche gradient. The rare taxa niche breadth indices significantly decreased along the niche gradient (*p* < 0.05), and the variety genotype was the main reason for the significant difference within the niche gradient (*p* < 0.05). The relationship between the population frequency and relative abundance was well described by the neutral community model (NCM) (Figure 4B), which showed that the rhizosphere bacterial community had the best fitted value (R^2^), followed by the endosphere bacterial community, and then the bulk soil bacterial community, accounting for 61.00%, 73.70%, and 69.00% of the bacterial community variance in the three niches, respectively. Thus, stochastic processes were more important in the rhizosphere bacterial community assembly process, and the contribution of the bulk soil bacterial community was lower. The Nm, which quantifies dispersal between communities, decreased along the niche gradient, indicating that proximity to the host roots limited the species dispersal (Figure 4B). The results of the analysis of abundant sub-communities revealed that the neutral model explained 69.40% and 70.70% of the variation in the moderate taxa and rare taxa in the rhizosphere, respectively, indicating that the rhizosphere bacterial community assembly process was determined by a strong stochastic process (Appendix A). In the endosphere, the variances of the moderate and rare taxa were 48.00% and 45.60%, indicating that the endosphere bacterial community assembly process was determined by specific competition and environmental filtering pressure (Appendix A). The Nm values of the moderate and rare taxa decreased along the niche gradient, and the Nm values of the moderate taxa were significantly higher than those of the rare taxa, indicating that the host plants limited the spread of the moderate taxa and, particularly, of the rare taxa.

The co-occurrence network of the three ecological niches of the two cotton varieties was constructed based on the Spearman correlation at the genus level (Figure 5A). The number of nodes and edges of the rhizosphere co-occurrence network was the highest, followed by the bulk soil, and then the endosphere. The average clustering coefficient, complexity index, and edge density, which characterize the stability of the co-occurrence network, were highest in the rhizosphere bacterial community, followed by the bulk soil, and then the endosphere, where all the differences were significant (*p* < 0.05) (Figure 5B). The degree centrality was significantly higher (*p* < 0.05) in the endosphere for the XLZ61 than the XLZ62, indicating that the variety genotype significantly impacted the species correspondence of the cotton endosphere bacterial community. The endosphere bacterial community showed the highest moderate centrality (Figure 5B) but the lowest robustness (Figure 5C), indicating that the natural connectivity of the root endophytic bacterial community was the highest and the community was more stable. The betweenness centralization of the endosphere bacterial community was the highest (Figure 5B), but the robustness was the lowest (Figure 5C), indicating that the rhizosphere bacterial community had the highest natural connectivity and community stability.

## 4. Discussion

The compartment niche is the main factor affecting the diversity of bacterial communities in the cotton soil–root continuum, and the cotton variety genotype is an important source of diversity in bacterial communities within the compartment niche. Our study found that the α-diversity of the rhizosphere bacterial community was significantly higher than that of the bulk soil, but the α-diversity of the endosphere bacterial community was significantly lower. We hypothesize that the continuous cropping system led to a large enrichment of the pathogenic bacteria in the bulk soil, which suppressed the relative abundance of beneficial microorganisms. This is mainly due to the “call for help” strategy of the cotton root system, which attracts beneficial microorganisms by releasing volatile organic compounds (VOCs) or altering the synthesis and secretion of specific root substances [49]. This is due to the fact that cotton roots recruit bacterial communities through the rhizosphere in the enrichment pattern to form a rhizosphere community dominated by moderate taxa, while the emergence of a large number of root-specific OTUs significantly increases the α-diversity index of the rhizosphere bacterial community [15]. The assembly process of the cotton endosphere bacterial community is filtered by the host plant root system [13,50]. More than half of the rhizosphere species failed to enter the cotton root endophytic environment, as the endosphere recruited abundant taxa with an accumulated relative abundance of 64.43–59.74% in the enrichment pattern, leading to a significant reduction in the α-diversity index of the endosphere communities, especially in the XLZ62. The variation patterns of the α-diversity indices of the three abundance sub-communities along ecological niches were completely different. We found that the variation in the α-diversity indices of the moderate taxa more accurately represents the total community, which is mainly affected by the rhizosphere deposition effect. Cotton genotype differences have a more significant impact on the diversity of rare taxa, and the endosphere shows a strong host selection effect with rare taxa. Therefore, we suggest that the moderate taxa contribute significantly more to changes in the bacterial community diversity than the rare and abundant taxa during the succession of bacterial communities in the cotton root–soil continuum.

Due to management measures such as the perennial continuous cropping and straw returning of Xinjiang cotton [34], the bulk soil bacterial community was mainly reduced to a few abundant taxa dominated by pathogenic bacteria, while a large number of beneficial bacteria were depleted to moderate taxa or even rare taxa, forming a skewed species abundance distribution [19]. Previous studies have suggested that plant roots can regulate the recruitment of beneficial microorganisms by root secretions through a “cry for help” mechanism [49], either directly through the production of antibacterial substances that inhibit pathogens or indirectly by utilizing induced systemic resistance to help plants resist pathogen invasion [51]. In support of this, we found that the cotton rhizospheres recruited a large number of antagonistic probiotics in an enrichment pattern, such as *Sphingomonas* from the phylum *Proteobacteria*, *Arthrobacter* from the phylum *Actinobacteria*, and *Bacillus* from the phylum *Firmicutes*, which may help improve the resistance of cotton to disease. At the same time, all bulk soil pathogenic bacteria migrated to the rhizosphere in a depletion pattern, reducing the damage to the cotton by soil pathogens. The recruitment process of the root endophytic bacterial community was dominated by a depletion pattern and supplemented by an enrichment pattern, which was regulated by the host genotype. We also found that the cotton endosphere mainly recruited many nitrogen-fixing, disease-resistant probiotics and ion absorption-related bacterial taxa in an enrichment pattern, including unidentified *Rhizobiaceae*, *Streptomyces*, *Ilumatobacter*, *Devosia*, *Dongia*, *Sphindomonas*, and *Pseudomonas*, to establish a symbiotic relationship with the cotton roots to promote healthy cotton growth. Our analysis revealed that the relative abundance of abundant taxa species was very high across the niche gradient, especially in the bulk soil and root endophytic bacterial communities. We found that the composition of abundant species dominated the biological functions of the niche, and the recruitment of abundant taxa in different niche gradients had a strong spatial specificity [52]. Therefore, different niche gradients along the cotton soil–root continuum in Xinjiang dominated the succession of community biological functions through the recruitment of differential abundant taxa. The combined effect of rhizosphere sedimentation and host selection during the bacterial community assembly process, with subsequent changes in the composition and relative abundance of the moderate and rare taxa, alters the compositional structure of bacterial communities and is the main reason for the segregation of host bacterial communities of different genotypes within niches.

Niche-based theories and neutral-based theories constitute two important and complementary mechanisms for understanding the assembly of the microbial community. We found that the niche breadth index of the cotton rhizosphere was significantly higher than that of the bulk soil and endosphere, while the neutral model theory explained the highest variance of the rhizosphere bacterial community. This indicated that root exudates and sediments provided ample, available resources for rhizosphere bacterial communities and that stochastic processes are more important in rhizosphere bacterial community assembly. The resources that the bulk soil and endosphere can provide for bacterial communities are limited; the proportion of deterministic processes increases in the process of community assembly, and the community has an obvious disadvantage in resource competition, resulting in an uneven species distribution [46]. At the same time, the closer to the host root system along the niche gradient, the more significant the diffusion restriction. We found that the niche breadth indices of the three abundance sub-communities were mainly affected by the niche gradient, and the variation pattern was consistent with the total community Shannon index, especially that of the moderate taxa. The genotype difference of cotton varieties within the niche is the main reason for the difference in the niche breadth index, especially in the endosphere, indicating that the genotype difference within the niche is the main reason for the differences in resource competitiveness. Previous studies have suggested that abundant taxa and rare taxa exhibit different community assembly patterns. This study showed that the assembly process of the moderate taxa and rare taxa is affected by strong stochastic processes in the cotton rhizosphere, while the assembly processes in the endosphere are subject to strong specific competition and ambient filtration pressure. Species dispersal between sub-communities becomes more restricted as it gets closer to the host root system (Figure 4C), and the rare taxa are more sensitive to these factors, which is consistent with previous studies [36,53].

We found that the co-occurrence networks of the bacterial communities in different niches of the cotton soil–root continuum were substantially different (Figure 5A). The rhizosphere bacterial community co-occurrence network had the largest number of nodes and edges, the highest network complexity, and a more stable bacterial community. However, it also showed a lower community modularity index, which may be related to the rhizosphere bacterial communities being dominated by moderate taxa. The root endophytic bacterial communities showed a significant decrease in species richness due to host selection effects, and the deterministic assembly process led to the fluctuation in the bacterial community species abundance of rich taxa, resulting in a decrease in the stability and complexity of the community co-occurrence network. We also found that the cotton variety genotype is an important factor in the topological differences in the endosphere bacterial community network.

## 5. Conclusions

In this study, we analyzed the effect of the soil–root niche gradient on the diversity of the bacterial community and the composition, assembly process, and co-occurrence network of the main cotton varieties grown in the northwest inland cotton region and found the following: (i) The bacterial communities dominated by pathogenic bacteria were formed in the cotton bulk soil under the influence of factors such as field management systems, while a large number of nitrogen-fixing and antibacterial species were depleted into rare taxa. The rhizosphere recruited antagonistic probiotics in an enrichment pattern driven by root exudates to form a bacterial community predominated by moderate taxa, which significantly improved the stability and resistance of the bacterial communities. The endosphere bacterial community assembly was dominated by depletion patterns, which filtered many rhizosphere bacterial species and formed an endosphere bacterial community dominated by antagonistic probiotics and nitrogen-fixing bacteria. (ii) The assembly process of the rhizosphere bacterial communities is mainly driven by stochastic processes, which have the highest niche breadth index and provide more resources such as rhizosphere exudates and sediments for bacterial community species competition to form abundant taxa-dominated rhizosphere bacterial communities. (iii) The assembly process of the bulk soil and endosphere bacterial communities was determined more by deterministic processes, which have a significantly reduced niche breadth index, and the competition for species resources is intensified. The host genotype regulation significantly reduces the α-diversity of the endosphere bacterial communities, restricts the community species competition, and reduces the stability and resistance of the co-occurrence network. This present study provides new insights into understanding the roles and contributions of the abundant sub-communities in the assembly of bacterial communities in the cotton soil–roots continuum. However, this study was mainly focused on the initial flowering stage of the cotton, so we conducted systematic research and analysis of the bacterial community assembly mechanism of different cotton growth and development stages in order to gain further understanding of the dynamic process of bacterial community assembly in different niche gradients.

## Figures and Tables

**Figure 1 microorganisms-12-00869-f001:**
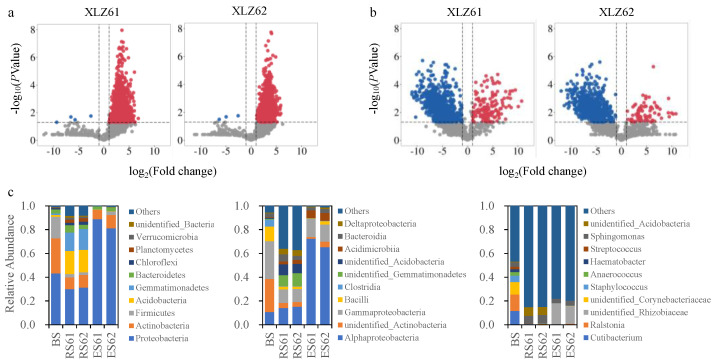
Analysis of the bacterial community composition and relative abundance change in different niches of the cotton soil–root continuum. (**a**) Volcano plots showing enrichment and depletion of OTUs in the relative abundance between the bulk soil (BS) and the rhizosphere (RS) bacterial communities of the XLZ61 and XLZ62. Colored dots indicate significant enrichment or depletion of OTUs compared with the bulk soil (*p* < 0.01). (**b**) Volcano plots showing enrichment and depletion of OTUs in the relative abundance between the RS and the endosphere (ES) bacterial communities of the XLZ61 and XLZ62. Colored dots indicate significant enrichment or depletion of OTUs compared with the bulk soil (*p* < 0.01). A volcanic plot of the changes in the OTU relative abundance between the RS and the endosphere (ES) bacterial communities of the XLZ61 and XLZ62. (**c**) A histogram of the relative abundance of the top 10 dominant taxa at the phylum, class, and genus levels of the bacterial communities in different compartment niches (BS, bulk soil; RS61, the rhizosphere of XLZ61; RS62, the rhizosphere of XLZ62; ES61, the en-dosphere of XLZ61; and ES62, the endosphere of XLZ6).

**Figure 2 microorganisms-12-00869-f002:**
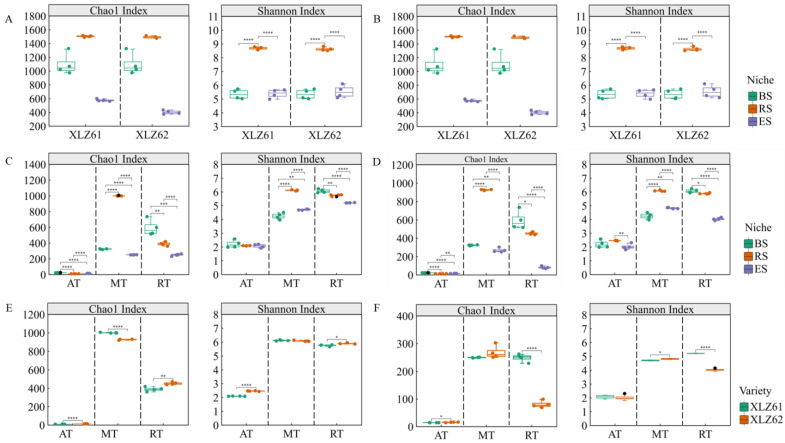
Bacterial community α-diversity analysis. (**A**) Chao1 index and Shannon box plot of bacterial communities among three ecological niches of two cotton cultivars. (**B**) Chao1 index and Shannon box plot of bacterial communities in the rhizosphere and endosphere of the two cotton cultivars. (**C**) Chao1 index and Shannon box plot of bacterial communities among three ecological niches of three sub-communities of XLZ61. (**D**) Chao1 index and Shannon box plot of bacterial communities among three ecological niches of three sub-communities of XLZ62. (**E**) Chao1 index and Shannon box plot between two cotton cultivars of three sub-communities in the rhizosphere. (**F**) Chao1 index and Shannon box plot between two cotton cultivars of three sub-communities in the endosphere. Note: wilcox rank sum test,* *p* < 0.05, ** *p* < 0.01, *** *p* < 0.001, **** *p* < 0.0001.

**Figure 3 microorganisms-12-00869-f003:**
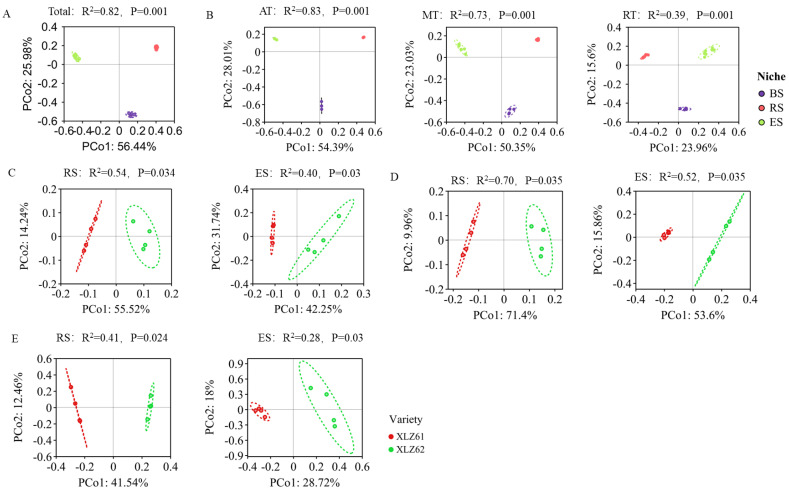
Bray–Curtis distance algorithm principal coordinate analysis (PCoA) of bacterial communities. (**A**) PCoA plot along the niche gradient. (**B**) PCoA plot of three sub-communities along the niche gradient. (**C**) PCoA plot of the two cotton varieties in the rhizosphere and endosphere. (**D**) PCoA plot of the moderate taxa of the two cotton varieties in the rhizosphere and endosphere. (**E**) PCoA plot of the rare taxa of the two cotton varieties in the rhizosphere and endosphere.

**Figure 4 microorganisms-12-00869-f004:**
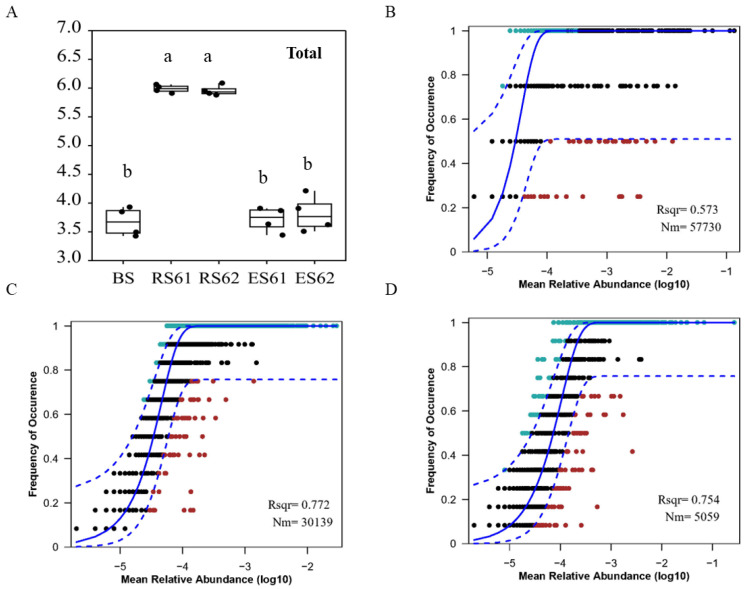
Niche breadth and neutral community model analysis among the niche gradients. (**A**) Levins’ niche breadth index analysis of the total community (BS, bulk soil; RS61, the rhizosphere of XLZ61; RS62, the rhizosphere of XLZ62; ES61, the endosphere of XLZ61; and ES62, the endosphere of XLZ6). Fit of the neutral community model (NCM) of the bulk soil (**B**), rhizosphere (**C**), and endosphere (**D**) community assembly, where the dashed blue lines represent 95% confidence intervals around the model prediction. OTUs that occur more or less frequently than predicted by the NCM are shown in different colors. Nm indicates the metacommunity size times immigration, and R^2^ indicates the fit to this model. Different letters in the same figure meant significant difference at 0.05 level.

**Figure 5 microorganisms-12-00869-f005:**
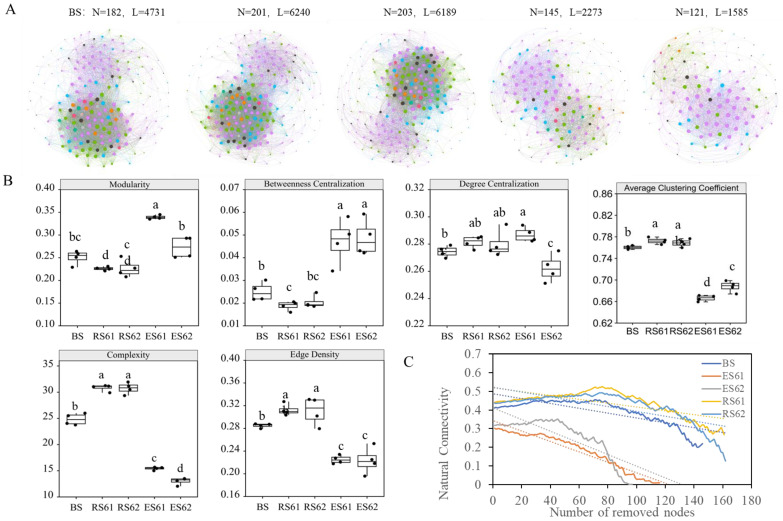
Network of co-occurring bacterial genera based on correlation analysis (r > 0.6, *p* < 0.05). A connection shows a strong and significant correlation. The size of each node is proportional to the node degree. (**A**) Co-occurring network of the bulk soil, rhizosphere, and endosphere of XLZ61 and XLZ62 (N: node number; and L: number of link edges). (**B**) Topology parameters of the co-occurrence network. (**C**) Robustness analysis. (Note the following: BS, bulk soil; RS61, the rhizosphere of XLZ61; RS62, the rhizosphere of XLZ62; ES61, the endosphere of XLZ61; and ES62, the endosphere of XLZ6). Different letters in the same figure meant significant difference at 0.05 level.

**Table 1 microorganisms-12-00869-t001:** Changes in the OTU number and relative abundance between two cotton varieties showing the assembly progress of bacterial communities.

Treatment	BS	RS		ES	
		XLZ61	XLZ62	XLZ61	XLZ62
Total OTUs	1428 (1)	1645 (1)	1660 (1)	787 (1)	611 (1)
AT OTUs	23 (66.31%)	9 (17.42%)	13 (21.08%)	14 (64.43%)	16 (59.74%)
MT OTUs	347 (30.26%)	1010 (79.25%)	941 (75.23%)	252 (34.1%)	347 (39.41%)
RT OTUs	1058 (3.43%)	626 (3.33%)	706 (3.69%)	521 (1.47%)	248 (0.85%)

Footnotes: BS, the bulk soil bacterial community; RS, the rhizosphere bacterial community; and ES, the endosphere bacterial community.

## Data Availability

The data presented in this study are deposited in the NCBI repository, with accession number PRJNA1085520.

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
