# Peer review of "The Contributions of Sub-Communities to the Assembly Process and Ecological Mechanisms of Bacterial Communities along the Cotton Soil–Root Continuum Niche Gradient"

_microorganisms, 2024, doi:10.3390/microorganisms12050869_

Round 1
Reviewer 1 Report
Comments and Suggestions for Authors
Please see attached file. Main concerns are that replication needs to be made clearer for all data collected and used. results and discussion section should be combined to increase interest. Additional comments in the file.

English reads very well. Very minor editorial suggestion attached. A further read through would be beneficial.
Reviewer 2 Report
Comments and Suggestions for Authors
Dear Editor
Many thanks for considering me a potential reviewer for the said article. This article is well structured and written, however, here are some queries and minor corrections (suggestions) that must be taken into consideration before the onward steps. My observations as follow;
Comments
1. Line 104, please cite this information ‘Allelochemicals secreted by the roots of crops in continuous cropping systems cause nutritional imbalance and self-toxicity’.
2. ‘Xinjiang’ please add Xinjiang (China). People in around the world are not aware of Xinjiang but know China well.
3. Line 297, please always mention two decimal numbers 73.7%...check whole manuscript.
4. Can authors organize Figure 4 like figure 3?…. Figure 4 is very difficult to read.
5. Please cite ‘The soil was collected at the initial flowering stage on July 2 using the “S” shaped random sampling method’ Line 477.
6. where Bj is the niche breadth, Line 545, please be careful.
7. in situ (line 114) must be italicized.
